# Polymer Brush Friction in Cylindrical Geometries

**Karel J. van der Weg, Guido C. Ritsema van Eck and Sissi de Beer *** 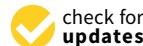

Materials Science and Technology of Polymers, MESA+ Institute for Nanotechnology, University of Twente, P.O. Box 217, 7500 AE Enschede, The Netherlands; k.j.vanderweg@student.utwente.nl (K.J.v.d.W.); g.c.ritsemavaneck@student.utwente.nl (G.C.R.v.E.)

* Correspondence: s.j.a.debeer@utwente.nl; Tel.: +31-53-489-3170

**Abstract:** Polymer brushes are outstanding lubricants that can strongly reduce wear and friction between surfaces in sliding motion. In recent decades, many researchers have put great effort in obtaining a clear understanding of the origin of the lubricating performance of these brushes. In particular, molecular dynamics simulations have been a key technique in this scientific journey. They have given us a microscopic interpretation of the tribo-mechanical response of brushes and have led to the prediction of their shear-thinning behavior, which has been shown to agree with experimental observations. However, most studies so far have focused on parallel plate geometries, while the brush-covered surfaces might be highly curved in many applications. Here, we present molecular dynamics simulations that are set up to study the friction for brushes grafted on the exterior of cylinders that are moving inside larger cylinders that bear brushes on their interior. Our simulations show that the density distributions for brushes on the interior or exterior of these cylinders are qualitatively different from the density profiles of brushes on flat surfaces. In agreement with theoretical predictions, we find that brushes on the exterior of cylinders display a more gradual decay, while brushes on the interior of cylinders becomes denser compared to flat substrates. When motion is imposed, the density profiles for cylinder-grafted brushes adapt qualitatively differently to the shear motion than observed for the parallel plate geometry: the zone where brushes overlap moves away from its equilibrium position. Surprisingly, and despite all these differences, we observe that the effective viscosity is independent of the radius of the brush-grafted cylinders. The reason for this is that the viscosity is determined by the overlap between the brushes, which turns out to be insensitive to the exact density profiles. Our results provide a microscopic interpretation of the friction mechanism for polymer brushes in cylindrical geometries and will aid the design of effective lubricants for these systems.

**Keywords:** polymer brush; friction; lubrication; tribology; molecular dynamics

## 1. Introduction

Biological lubricants, such as those found in mammalian joints, outperform most industrial lubricants in efficiency and durability [1]. These excellent lubricating properties arise due to a complex mix of mechanisms [2]. One particularly important mechanism is that hydrophilic sugar chains attach to proteins and cartilage tissue to form brush-like structures [2,3]. These structures absorb low-viscosity aqueous liquids and thereby create a soft, slippery interface. This effect can be mimicked by end-anchoring polymers to surfaces to form so-called polymer brushes [4]. With these brushes, effective and durable lubricants have been developed that can work under aqueous [5,6] as well as oil environments [7].

Polymer brushes consist of long macromolecules that are attached with one end [8] or both ends [9] to a surface at a density that is high enough such that the polymers stretch away from

the grafting surface. When immersed in good solvents, these brushes swell and absorb the solvent. Consequently, the solvent will remain in the contact area [10], even at normal loads that are up to tens of MPa [11]. This results in low friction upon relative sliding of the brush-covered surfaces [4,11]. At higher compressions, opposing brushes can interdigitate [12,13], which can increase frictional stresses by several orders of magnitude [14]. In fact, interdigitation between brushes can be switched using asymmetric brushes [15] and this can be employed to control the friction between surfaces [16].

Molecular simulations have been employed extensively to obtain a microscopic picture of polymer brush friction [17–26]. With help of these simulations, friction-velocity relations have been derived [27–32]. For bilayer brush-systems, shear stresses act in the overlap zone (pink central region in Figure 1a) where polymers from the opposing brushes interact. At low velocities, the brushes are in thermal equilibrium and friction arises due to viscous dissipation. Under these conditions, friction is expected to increase linearly with velocity. At shear rates that are higher than the inverse characteristic relaxation times within the brush, the polymers are dragged along the shear direction. Therefore, they tilt and stretch (see Figure 1a), which reduces the overlap length $l$. Due to the reduction in overlap with increasing velocity, the friction force $f$ increases sub-linearly with velocity $v$ at high shear rates, i.e., $f \propto v^\kappa$. The shear-thinning exponent $\kappa$ depends on the compression as well as the nature of the polymers and varies between 0 and almost 1 [27–32]. For symmetric, solvated, neutral, linear bilayer brushes, the shear-thinning exponent is expected to be $\kappa = 0.52$–$0.58$ [31].

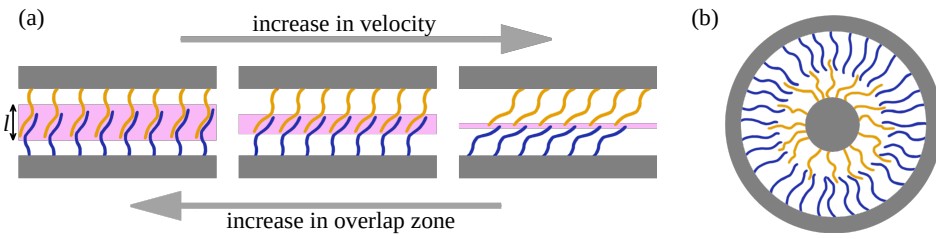

**Figure 1.** (**a**) Schematic representation of the shear-induced reduction of the overlap length $l$ for increasing shear velocities (**b**) The geometry employed in our simulations. It consists of an outer cylinder that has polymers grafted on the inner surface and an inner cylinder that has polymers grafted on the outer surface. The cylinders will move in opposite directions in and out of the drawing plane.

Most theoretical and simulation studies of polymer brush friction focus on parallel plate geometries [17–29,31,32]. However, the contact geometries might be different in applications where brushes are applied. For example, the contact area will have a cylindrical shape for brush-coated fibers, (nano-)tubes and (nano-)needles. In medical applications needle injection accuracy can be improved if the contact friction can be reduced using polymer brush coatings. Another example where cylindrical geometries are relevant is in the nuclear pore complex (NPC), which is often modeled as a polymer brush on the inside of a cylinder. If brushed particles move through these NPCs, they will experience friction through a cylindrical geometry [33]. In particular, in the latter systems, curvatures can be high. On such curved surfaces, the density profiles of brushes can be different from that on flat surfaces [34]. For brushes on flat surfaces, the density profile has, approximately, a parabolic shape [35,36]. In contrast, brushes on the exterior of cylinders display a shallower decay in the density, which can become concave for small cylinder radii $R$ [37,38]. Brushes grafted on the interior of cylinders can be stretched or compressed depending on the chain length, grafting density and $R$ [39–41]. Due to these deviating density profiles, one can expect that the interaction of a brush on the *interior* of a cylinder that is in contact with a brush on the *exterior* of a cylinder (see Figure 1b) is different from that of a parallel plate geometry too. Therefore, the frictional properties for these geometries can be expected to be different as well. Yet, to our knowledge, this research topic has not been explored so far.

In this article, we report the results of molecular dynamics simulations in which we study the friction-velocity relations for neutral bilayer brushes in cylindrical geometries. Our results show that the density distributions of brushes significantly alter as the cylinder radius becomes smaller.

Surprisingly, however, the lubricating properties are unaltered, and the effective viscosities are found to be independent of the curvature of the grafting surface.

## 2. Models and Methods

Our simulation cells consist of two cylinders with different radii (see Figure 2), $R_{\text{in}}$ for the inner cylinder and $R_{\text{out}}$ for the outer cylinder. The outer cylinder has polymers grafted on the inner surface and the inner cylinder has polymers grafted on the outer surface. The polymers are represented by beads connected by springs (Kremer–Grest model [42]). This model is well known to qualitatively describe the tribo-mechanical properties of surface-attached polymers [31]. Within this model, non-consecutive beads interact via the Lennard–Jones potential:

$$U_{\text{LJ}} = 4\epsilon \left[ \left( \frac{\sigma}{r} \right)^{12} - \left( \frac{\sigma}{r} \right)^{6} - \left( \frac{\sigma}{r_{\text{c}}} \right)^{12} + \left( \frac{\sigma}{r_{\text{c}}} \right)^{6} \right], \tag{1}$$

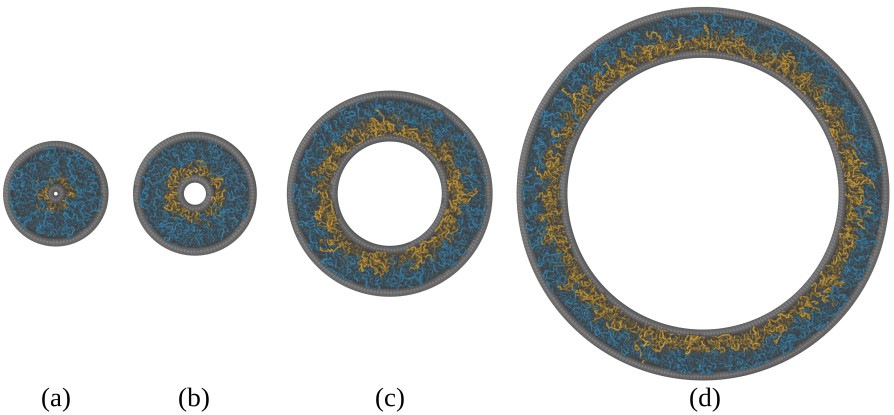

(a)　　　　　(b)　　　　　(c)　　　　　　　　(d)

**Figure 2.** Simulation snapshots of the models used with radii (**a**) $R_{\text{in}} = 2\sigma$, (**b**) $R_{\text{in}} = 5\sigma$, (**c**) $R_{\text{in}} = 20\sigma$, (**d**) $R_{\text{in}} = 50\sigma$. Snapshots are rendered in VMD [43].

where $\epsilon$ equals the depth of the potential well and represents the unit of energy ($\epsilon = 1$), $\sigma$ is the zero-crossing distance and represents the unit of distance ($\sigma = 1$). The distance between the beads is given by $r$ and $r_{\text{c}}$ is the length beyond which the potential is set to zero. Our interactions are made purely repulsive by truncating the potential at $r_{\text{c}} = 2^{1/6}\sigma$. Through this repulsion good solvent conditions are mimicked. Solvent-free systems such as this and explicit solvent models are identical in the limit of high grafting density through constraints on the system's total particle density. Results by Galuschko et al. [27] indicate that the two do indeed display the same scalings despite ostensibly different behavior at finite grafting densities. This requires the preservation of hydrodynamic relations, however. This was achieved through our choice of thermostat. Consecutive beads interact via the finite extensible non-linear elastic (FENE) potential:

$$U_{\text{FENE}} = -0.5KR_0^2 \ln \left[ 1 - \left( \frac{r}{R_0} \right)^2 \right], \tag{2}$$

combined with the repulsive part of the Lennard–Jones potential $U_{\text{LJ}}$. The stiffness $K$ is set to $K = 30\epsilon/\sigma^2$ and maximum extension $R_0 = 1.5\sigma$, which ensures that no bond-crossing can occur [42]. A polymer bead represents a Kuhn unit, which has a size and interaction that depends on the type of polymer. For example, for polyethylene, the unit of energy $[\epsilon]$ equals 30 meV [44]. Moreover, one bead may represent 3–4 monomers, such that $\sigma$ equals 0.5 nm. The unit of mass is $[m] = 10^{-22}$ kg and, since

we employ Lennard–Jones units, $m$ is set to 1 in the simulations (for all beads). The unit of velocity is $[v]$ and is controlled by the simulation.

We have set up eight cylinder-systems of different sizes and grafting densities (see Table 1). The system-sizes vary between approximately $R_{in} = 2\sigma$ and $R_{in} = 50\sigma$ (see Figure 2). The cylinders are effectively infinite by application of periodic boundary conditions in the direction of the cylinder axis. The box lengths in this direction are 44 and 69 $\sigma$ for the low- and high-density brushes, respectively. The wall-to-wall distance $D$ (15.44 and 15.29 $\sigma$) and grafting densities $\alpha_g$ (0.085 and 0.17 $\sigma^{-2}$) have been chosen to be close to values employed in Ref [27], such that our results can be compared to their results for the parallel plate geometry. Our surface coverage is small enough to prevent crowding effects [41]. Yet, it is high enough to ensure we are in the brush regime (2.65x and 5.3x the critical grafting density for brush formation) [45]. The cylinder walls are built up of a single layer of atoms in a triangular lattice using a lattice constant of 1.2 $\sigma$. The wall atoms are fixed on their lattice sites. The polymers consist of $N = 30$ repeat units, which is long enough to mimic the tribo-mechanical properties of polymer brushes on a qualitative level [14]. The polymers are anchored at their chain-ends to wall atoms via the Lennard–Jones potential ($\epsilon = 100$, $\sigma = 1.8$ and $r_c = 2.5\sigma$). The interactions between all other polymer beads and the wall is also described by the Lennard–Jones potential and made purely repulsive by positioning the cut-off in the potential minimum ($\epsilon = 1$, $\sigma = 1$ and $r_c = 2^{1/6}\sigma$). Because of these repulsive interactions, we do not suffer from a wall-induced brush collapse [46]. The models used are in the scale of nano-applications, in this high curvature regime we expect to see the most change in brush behavior versus the radius of the cylinder-systems. However, we predict that for most larger applications the curvature effect would be negligible and we find a limit on radius at which the brush would effectively behave as if grafted on a flat plate.

**Table 1.** Overview of the (geometric) parameters for our simulation systems. It gives the inner radius $R_{in}$, the outer radius $R_{out}$, the wall - to - wall distance $D$ and the grafting density $\alpha_g$.

| System | $R_{in}$ $[\sigma]$ | $R_{out}$ $[\sigma]$ | $D$ $[\sigma]$ | $\alpha_g$ $[\sigma^{-2}]$ |
|---|---|---|---|---|
| 1 | 2.3 2 | 17.76 | 15.44 | 0.085 |
| 2 | 5.40 | 20.84 | 15.44 | 0.085 |
| 3 | 20.07 | 35.51 | 15.44 | 0.085 |
| 4 | 50.18 | 65.62 | 15.44 | 0.085 |
| 5 | 2.18 | 17.47 | 15.29 | 0.170 |
| 6 | 5.46 | 20.75 | 15.29 | 0.170 |
| 7 | 20.20 | 35.49 | 15.29 | 0.170 |
| 8 | 50.22 | 65.51 | 15.29 | 0.170 |

The velocities and positions are updated using the velocity Verlet algorithm, as implemented in LAMMPS [47], using a time step of $\Delta t = 0.005\tau$, where $\tau = \sqrt{\epsilon/m\sigma^2}$. The simulations are performed in the $NVT$ ensemble. The temperature is kept constant at $T = 1.68\epsilon/k_B$ using a Langevin thermostat (damping coefficient $\zeta = 1\tau^{-1}$), which is switched off in the direction of motion to prevent potential artifacts [48]. This has been shown to produce functionally identical results to dissipative particle dynamics (DPD) thermostatting, which satisfies local conservation of momentum [48]. Sliding motion is imposed on the wall atoms in opposite directions for the two cylinders ($v$ and $-v$) parallel to the axis of the cylinder and the friction force is calculated from the average force on the wall atoms [49].

To obtain a microscopic interpretation of the origin of the friction forces, we study the density profiles $\rho(r)$ of the polymer brushes. To extract them from our simulations, we split up the box in small volumes in the radial direction using a grid length $\Delta r = 0.25\sigma$. We monitor the average number of particles $N_p$ in these volumes $V_r$. Due to the cylindrical domain decomposition, the volumes change as a function of $r$. We compensate for this effect in our calculations of the density profile $\rho(r) = N_p(r)/V_r(r)$. As common in brush friction simulations [14,16,18,19,30], we compute the

overlap integral to quantify the degree of overlap between the opposing brushes. The overlap integral is defined as:

$$I_{ov} = \int_{R_{in}}^{R_{out}} \rho_{in}(r)\rho_{out}(r)dr, \tag{3}$$

where $\rho_{in}(r)$ and $\rho_{out}(r)$ are the density profiles of the brushes grafted on the inner and outer cylinder walls, respectively.

## 3. Results and Discussion

### 3.1. Density Profiles

#### 3.1.1. Brushes on Individual Cylinders

To illustrate the effect of cylinder curvature on the density distribution of polymer brushes, we compare the polymer partitioning for brushes grafted on the interior or exterior of cylinder walls without the opposing brush present. Figure 3 shows the density profiles $\rho_{in}(r)$ and $\rho_{out}(r)$ for cylinders of different radii $R$ that are coated either on the outside of an inner cylinder of radius $R_{in}$ (Figure 3a,c) or the inside of an outer cylinder of radius $R_{out}$ (Figure 3b,d). The top two graphs give the density profiles for low grafting density brushes ($\alpha_g = 0.085\sigma^{-2}$) and the bottom two graphs give the density profiles for high grafting density brushes ($\alpha_g = 0.170\sigma^{-2}$). In these graphs the position of the wall is set to be $r = 0$ and $\overrightarrow{r}$ and $\overleftarrow{r}$ represent the radial distance from the wall in the positive or negative direction, respectively. For comparison, the density profiles of brushes grafted on flat surfaces have been included in the graphs (orange stars).

The graphs in Figure 3 show that there is a clear effect of the wall-curvature on the density profiles. Figure 3a gives the density profiles for the low grafting density brushes ($\alpha_g = 0.085 \sigma^{-2}$) grafted on the exterior of inner cylinders with radii between $R_{in} = 2\sigma$ and $R_{in} = 50\sigma$ and the density profile of a brush of the same grafting density attached to a flat wall ($R_{in} \rightarrow \infty$). Close to the wall, we observe sharp oscillations in the density. These oscillations are commonly observed in molecular simulations of polymer brushes [50–52] and arise due to the wall constraint, which gives rise to a layering effect. The onset of layering depends on the radius of the cylinder, which could be an indication of a deadzone, which has been predicted theoretically [53]. Yet, it could also be caused by the lower effective bonding strength experienced by anchor molecules at high curvatures such that they make less intimate contact with the wall. The density distribution for the flat wall shows the characteristic parabolic decay as a function of the distance from the surface that is well-known to occur for brushes, in particular in good solvents [35,36,54]. Moreover, we observe a rounded gradual density decay near the top of the brush at $\overrightarrow{r} \approx 15\sigma$. This decay can be attributed to thermal fluctuations in the concentration at the outer region [55], but is also caused by the translational entropy of the polymer ends [56].

The density profiles in Figure 3a for the cylindrical walls become increasingly less convex as the radius of the cylinder decreases. For the smallest two cylinders ($R_{in} = 2\sigma$ and $R_{in} = 5\sigma$) the density decays approximately linearly as a function of the distance from the surface. In the limit of small $R_{in}$ and low grafting densities, theory predicts that the density profiles can even become concave [57]. We do not reach this limit. The overall density for the profiles in Figure 3a also decreases when $R_{in}$ decreases. The reason for this is the strong increase in volume(-availability) as a function of $\overrightarrow{r}$ for small $R_{in}$. All these observations are consistent with earlier reports of MD simulations of brushes on the exterior of cylinders [37].

Figure 3c shows the density profiles for the high grafting density brushes ($\alpha_g = 0.170\sigma^{-2}$) grafted from the exterior on an inner cylinder of various radii $R_{in}$ as well as a flat surface ($R_{in} \rightarrow \infty$). The density profile of the brush on the flat surface (orange stars) displays a flatter distribution than the parabolic distribution for our low grafting density brush. This is to be expected for high-density brushes [52,58,59] and caused by saturation effects at high densities. The density profiles of the high-density brushes grafted on the cylinders show the same trend as the low-density brushes. Upon

decreasing $R_{\text{in}}$, the density profiles become increasingly less convex and for $R_{\text{in}} = 2\sigma$ the density decays almost linearly. Moreover, the overall density becomes also here smaller for smaller $R_{\text{in}}$.

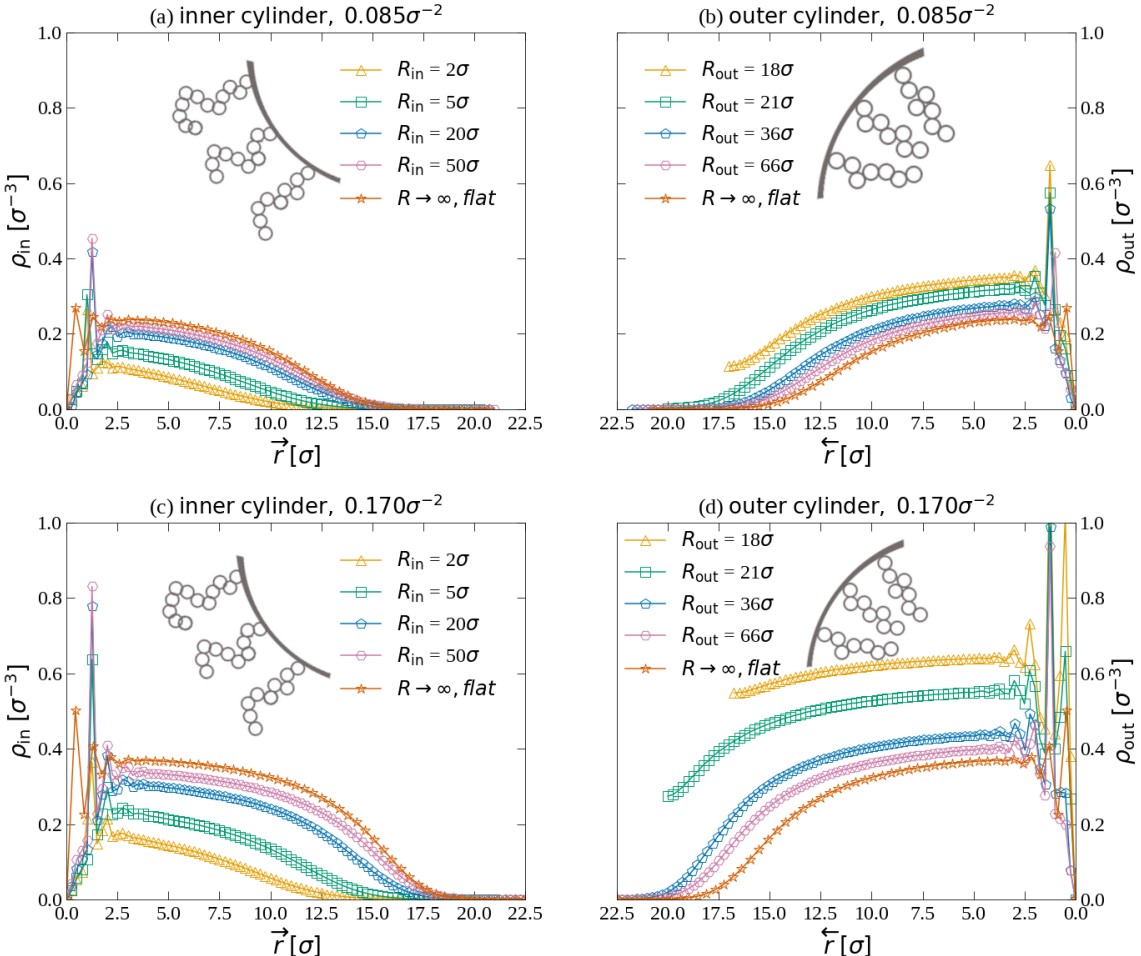

**Figure 3.** The density profiles for brushes grafted on the interior or exterior of individual cylinders of different radii $R$ as well as for brushes grafted from flat surfaces ($R \rightarrow \infty$). (**a**) The brush density $\rho_{\text{in}}$ as a function of the distance from the surface $\overrightarrow{r}$ for grafting density $\alpha_g = 0.085\,\sigma^{-2}$. The brushes are grafted on the exterior of an inner cylinder of radius $R_{\text{in}}$. (**b**) the brush density $\rho_{\text{out}}$ as a function of $\overleftarrow{r}$ for $\alpha_g = 0.085\,\sigma^{-2}$. The brushes are grafted on the interior of an outer cylinder of radius $R_{\text{out}}$. (**c**) $\rho_{\text{in}}$ as a function of $\overrightarrow{r}$ for $\alpha_g = 0.170\,\sigma^{-2}$. The brushes are grafted on the exterior of an inner cylinder of radius $R_{\text{in}}$. (**d**) $\rho_{\text{out}}$ as a function of $\overleftarrow{r}$ for $\alpha_g = 0.170\,\sigma^{-2}$. The brushes are grafted on the interior of an outer cylinder of radius $R_{\text{out}}$.

Figure 3b displays the density distributions for the low grafting density brushes ($\alpha_g = 0.085\sigma^{-2}$) grafted on the interior of the outer cylinders with radii between $R_{\text{out}} = 18\sigma$ until $R_{\text{out}} = 66\sigma$, without the inner cylinder being present. For large radii, the density profiles are comparable to the density profile for the brush on the flat surface ($R_{\text{out}} \rightarrow \infty$) and show a characteristic parabolic decay as a function of the distance from the surface. Upon decreasing the cylinder radius, the total density of the brush increases. This increase is also observed when one increases the polymer length $N$, while keeping the radius constant [40] and is caused by a crowding effect. For the smallest cylinder radius ($R_{\text{out}} = 18\sigma$), the density does not decay to zero for large $\overleftarrow{r}$. The reason for this is that opposing polymers on the interior of the cylinder start to interact with each other for these small radii. This also is consistent with earlier simulation results [40,41].

In Figure 3d we show the density profiles from high-density brushes ($\alpha_g = 0.170\sigma^{-2}$) grafted on the interior of cylinders of different radii between $R_{\text{out}} = 18\sigma$ until $R_{\text{out}} = 66\sigma$, without the presence

of an inner cylinder. Due to the higher grafting density and, thereby, stronger crowding effects, the overall brush density increases more strongly than in Figure 3b. For the same reason, the gradual decay to zero at large $\overleftarrow{r}$ disappears already for larger $R_{out}$ compared to the lower grafting density brush (Figure 3b).

To compare our simulation results on the density profiles to theoretical predictions, we determine heights $H_r$ of the different brushes. Different methods have been employed to determine the height of a polymer brush [14,17,18,34,37,40]. We choose to extract the brush height via the inflection point of the density profiles. The results are shown in Figure 4. The heights of brushes grafted on the exterior of the inner cylinders (orange triangles and squares) in Figure 4 show that the height $H_r \propto R^{1/4}$ for small radii $R$. This dependency is in exact agreement with the theoretical prediction by Wijmans and Zhulina [38] that:

$$H_r \propto \left( R^{d-1} \right)^{1/(d+2)} \text{ for } R \ll N, \tag{4}$$

where $d$ is the dimensionality of the system. For our cylinder-geometry $d = 2$, such that $H_r \propto R^{1/4}$ and this is indeed what we observe. For large radii the brush height approaches a constant value, which equals the height of brushes grafted on flat surfaces: for $\alpha_g = 0.085\sigma^{-2}$ this is $12.1\sigma$ and for $\alpha_g = 0.170\sigma^{-2}$ this is $16.0\sigma$.

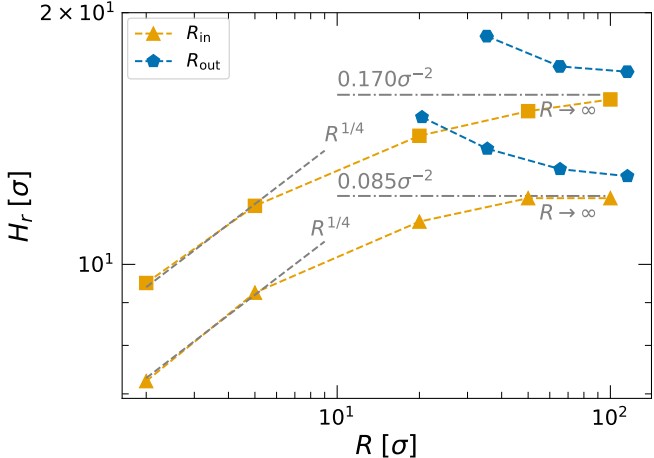

**Figure 4.** The brush height $H_r$ as a function of the cylinder radius $R$ for brushes grafted on the exterior of an inner cylinder of radius $R_{in}$ (orange squares for $\alpha_g = 0.170\sigma^{-2}$ and triangles for $\alpha_g = 0.085\sigma^{-2}$) and for brushes grafted in the interior of outer cylinders of radius $R_{out}$ (blue hexagons for $\alpha_g = 0.170\sigma^{-2}$ and pentagons for $\alpha_g = 0.085\sigma^{-2}$). The gray dashed lines with the theoretical prediction that $H_r \propto R^{1/4}$ for brushes on the exterior of cylinders. The gray dash-dotted lines denote the heights for brushes grafted on flat surfaces at grafting densities $\alpha_g = 0.170\sigma^{-2}$ (top) and $\alpha_g = 0.085\sigma^{-2}$ (bottom).

The scaling relation between the brush height $H_r$ and the cylinder radius for brushes grafted on the interior of cylinders with radii $R_{out}$ (blue hexagons and pentagons in Figure 4) is more difficult to confirm. The reason for this is that there is no smooth decay to zero at large $\overleftarrow{r}$ for cylinders of very small $R_{out}$ due to the interaction between the opposing polymers (see Figure 3b,d). Binder and Milchev [34] predict using scaling arguments that $H_r \propto R^{-1/4}$ for small $R$. Our data show indeed that the height decreases sub-linearly as a function of $R$ for small $R$. Yet, we cannot determine the exact exponent of the power law, as our results already leave the regime for high curvature and approach the regime in which the height is constant and equal to the height for $R \to \infty$. Due to the present representation employed in Figure 4, it appears that the heights at $R = 116\sigma$ (blue pentagons and hexagons) are still far from the height at $R \to \infty$. However, the remaining height difference is only $1\sigma$, which just appears large due to the logarithmic scale of the height axis.

### 3.1.2. Bilayer Brushes between Two Cylinders

Now that we have established the polymer partitioning for brushes grafted on the exterior or interior of cylinders, we will turn our attention to the density distributions for bilayer brushes confined between two cylinders. Figure 5 shows the density of brushes grafted on the exterior of inner cylinders of radii between $R_{in} = 2 - 50\sigma$ that are in contact with brushes grafted on the interior of outer cylinders. The distance between the cylinder walls is kept constant at $D = 15.44\sigma$ for $\alpha_g = 0.085\sigma^{-2}$ and at $D = 15.29\sigma$ for $\alpha_g = 0.170\sigma^{-2}$. An overview of the studied systems is given in Table 1. For reference, we plot the density profiles for brushes between flat surfaces as well.

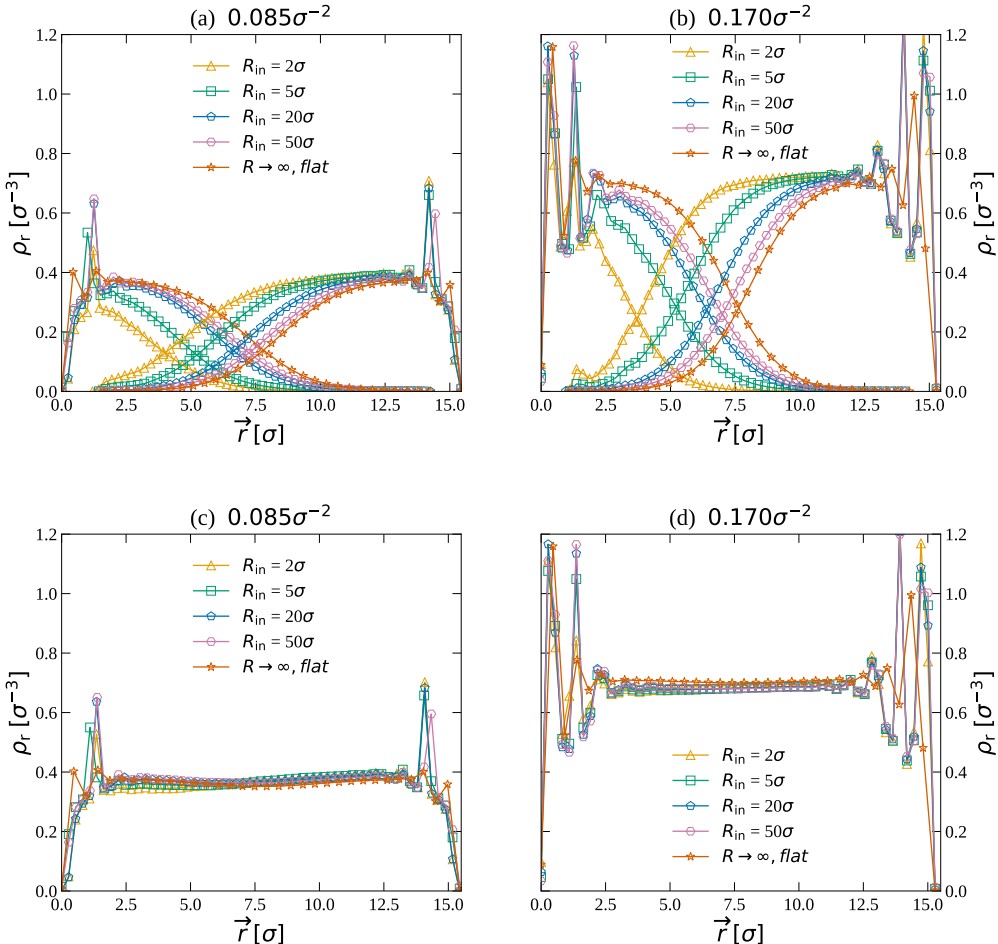

**Figure 5.** The density profiles for brushes grafted on the interior and exterior of the cylinder models of different radii $R$ as well as for brushes grafted from flat surfaces ($R \rightarrow \infty$). The brushes are grafted on the exterior of the inner wall and the exterior of the outer wall. The inner cylinder of radius $R_{in}$ is shown in the legend where the outer cylinder is of radius $R_{in} + D$. $D = 15.44$ for $\alpha_g = 0.085\sigma^{-2}$ and $D = 15.28$ for $\alpha_g = 0.170\sigma^{-2}$. (**a**) The brush density $\rho_{in}$ (left curve) and $\rho_{out}$ (right curve) as a function of the distance from the surface $\overrightarrow{r}$ for a grafting density of $\alpha_g = 0.085\ \sigma^{-2}$. (**b**) The brush density $\rho_{in}$ (left curve) and $\rho_{out}$ (right curve) as a function of the distance from the surface $\overrightarrow{r}$ for a grafting density of $\alpha_g = 0.170\ \sigma^{-2}$. (**c**) The total monomer density $\rho_{in} + \rho_{out}$ as a function of the distance from the surface $\overrightarrow{r}$ for a grafting density of $\alpha_g = 0.085\ \sigma^{-2}$. (**d**) The total monomer density $\rho_{in} + \rho_{out}$ as a function of the distance from the surface $\overrightarrow{r}$ for a grafting density of $\alpha_g = 0.170\ \sigma^{-2}$.

In Figure 5a we show the density profiles for the low grafting density brushes ($\alpha_g = 0.085 \, \sigma^{-2}$). The density profiles for the largest cylinder ($R_{\text{in}} = 50\sigma$) are close to the density profiles for $R \to \infty$, except that the position of the overlap zone has moved slightly towards the inner cylinder. This can be attributed to the asymmetry in the density profiles: For brushes on the exterior of inner cylinders, the density is smaller than the density for $R \to \infty$. In contrast, the density for brushes on the interior of outer cylinders is higher than the density for $R \to \infty$. When the two brushes are in contact, the higher density of the brush on the outer cylinder pushes the brush on the inner cylinder further inwards. Therefore, the overlap zones are moved towards the inner cylinder. As is clearly visible in Figure 5, this effect is stronger for smaller cylinder radii. While the shift in the overlap zone is less than $1\sigma$ for $R_{\text{in}} = 50\sigma$, it is more than $3.5\sigma$ for $R_{\text{in}} = 2\sigma$. The effect is even stronger for the high-density brushes (Figure 5b, $\alpha_g = 0.170 \, \sigma^{-2}$). For these brushes, the shift in the overlap zone is more than $1\sigma$ for $R_{\text{in}} = 50\sigma$ and more than $4\sigma$ for $R_{\text{in}} = 2\sigma$.

In Figure 5c,d we show the total density between the cylinders for the different cylinder radii. The profiles show that the total density between the cylinders is independent of the cylinder radii and only depends on the grafting density. For $\alpha_g = 0.085 \, \sigma^{-2}$, the average density $\rho \approx 0.39\sigma^{-3}$ and for $\alpha_g = 0.085 \, \sigma^{-2}$, the average density $\rho \approx 0.68\sigma^{-3}$. A constant density that is independent of the cylinder radius is to be expected, since the surface area of the cylinders increases linearly with their radius such that the total area and the volume between the walls stay the same relative to each other. This might seem counter-intuitive, considering that this relation for only either the inner or outer cylinder is non-linear. Yet, combined the average density is constant [41].

### 3.1.3. Bilayer Brushes between Two Cylinders in Relative Motion

In this section, we will study the effect of shear motion on the density distributions of the brushes. To do so, we move the inner cylinder with velocity $v$ in the direction of the cylinder axis, while the outer cylinder is moved in the opposite direction with velocity $-v$. Figure 6 shows the density profiles for brushes grafted on cylinders with small radii ($R_{\text{in}} = 5\sigma$, Figure 6a,b) and large radii ($R_{\text{in}} = 50\sigma$, Figure 6c,d) for no shear motion, for $\dot{\gamma} = 1 * 10^{-2}\tau^{-1}$ and $\dot{\gamma} = 1 * 10^{-1}\tau^{-1}$. In Figure 6b shear rates of $\dot{\gamma} = 1 * 10^{-3}\tau^{-1}$ and $\dot{\gamma} = 1 * 10^{-2}\tau^{-1}$ are shown, as for the high shear rate of $\dot{\gamma} = 1 * 10^{-1}\tau^{-1}$ the shear stresses at the anchor points become too high and the polymers start to hop over the surface. The density profiles for the brushes grafted onto the cylinder with large radii (Figure 6c,d) alter upon increasing the shear rate in a way that is very similar to the changes in density profiles for bilayer brushes between flat surfaces [27,28,31]. For small shear rates, the density profiles are the almost same as the density profiles without shear motion (see Figure 5). At these low shear rates, we move slow enough compared to the relaxation time of the polymers and the system is in thermal equilibrium. In contrast, when the imposed motion is faster than the relaxation time of the polymers, the density profiles clearly change. Upon increasing the shear rate, the density near the wall increases and the decay near the interface between the brushes becomes much sharper. Moreover, the overlap between the density profiles also clearly decreases. The overlap zone slightly shifts towards the center of the contact. However, this effect is still small.

The change in the density profiles for brushes grafted on cylinders of small radii, as shown in Figure 6a,b are qualitatively different from the changes we observe for the large radii in Figure 6c,d and those for the brushes on flat surfaces reported in the literature [27,28,31]. As the brushes on flat surfaces, the brushes on these highly curved cylinders become denser near the wall and display a sharper decay resulting in a smaller overlap between the brushes. However, on top of that, the position of the overlap zone shifts towards the center upon increasing the shear rate. This shift is approximately $1\sigma$.

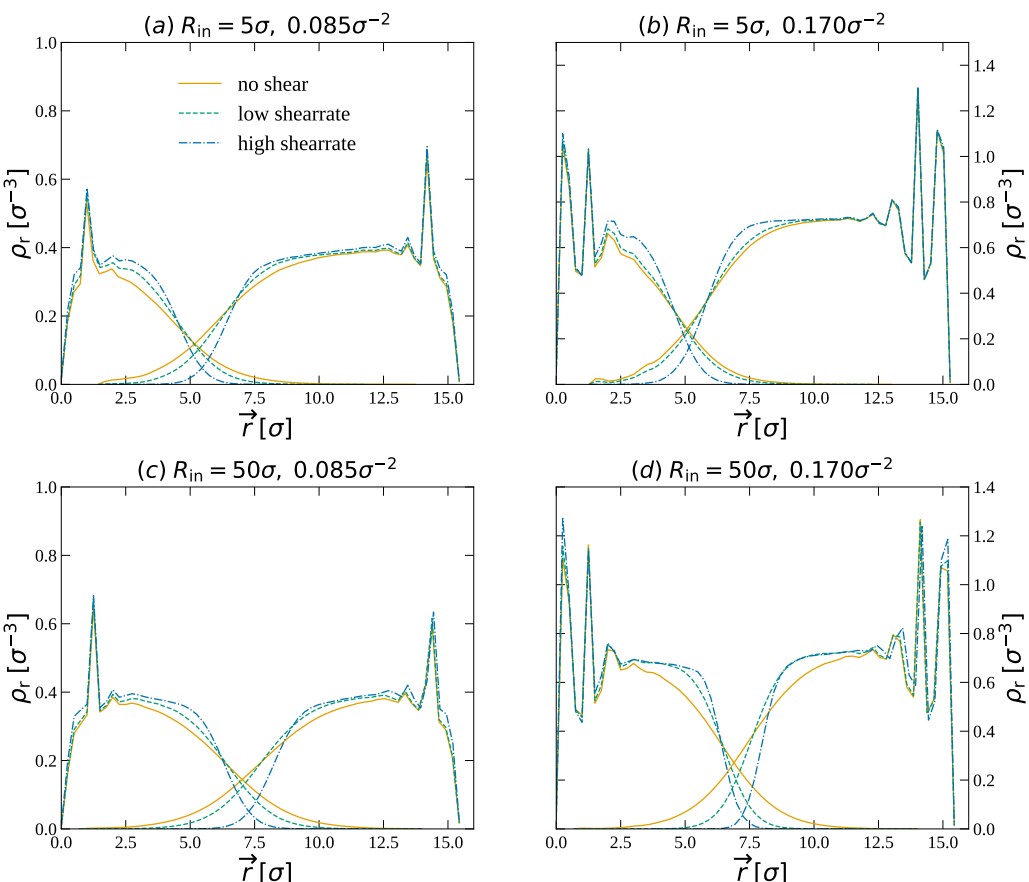

**Figure 6.** The density profiles for brushes grafted on the interior and exterior of the cylinder models of different radii $R$ for no shear motion, for $\dot{\gamma} = 1 * 10^{-2} \tau^{-1}$ and $\dot{\gamma} = 1 * 10^{-1} \tau^{-1}$. The brushes are grafted on the exterior of the inner wall and the exterior of the outer wall. In (**b**) shear rates of $\dot{\gamma} = 1 * 10^{-3} \tau^{-1}$ and $\dot{\gamma} = 1 * 10^{-2} \tau^{-1}$ are shown, as for the high shear rate of $\dot{\gamma} = 1 * 10^{-1} \tau^{-1}$ the shear stresses at the anchor points become too high and the polymers start to hop over the surface. (**a**) The brush density $\rho_{in}$ (left curve) and $\rho_{out}$ (right curve) as a function of the distance from the surface $\vec{r}$ for a grafting density of $\alpha_g = 0.085 \ \sigma^{-2}$ and cylinder radius $R_{in} = 5\sigma$. (**b**) The brush density $\rho_{in}$ (left curve) and $\rho_{out}$ (right curve) as a function of the distance from the surface $\vec{r}$ for a grafting density of $\alpha_g = 0.170 \ \sigma^{-2}$ and cylinder radius $R_{in} = 5\sigma$. (**c**) The brush density $\rho_{in}$ (left curve) and $\rho_{out}$ (right curve) as a function of the distance from the surface $\vec{r}$ for a grafting density of $\alpha_g = 0.085 \ \sigma^{-2}$ and cylinder radius $R_{in} = 50\sigma$. (**d**) The brush density $\rho_{in}$ (left curve) and $\rho_{out}$ (right curve) as a function of the distance from the surface $\vec{r}$ for a grafting density of $\alpha_g = 0.170 \ \sigma^{-2}$ and cylinder radius $R_{in} = 50\sigma$.

To further explain the shear-induced changes in the density profiles, the chain extension of the polymers, $R_g^2 / R_{g0}^2$ as a function of the shear rate is shown in Figure 7. Here $R_g^2$ is the average radius of gyration of all the polymers coated to a single wall. This is scaled by the radius of gyration without shear motion $R_{g0}^2$. The radius of gyration is calculated by the LAMMPS simulation software, using the following formula:

$$R_g^2 = \frac{1}{M} \sum_i m_i (r_i - r_{cm})^2, \tag{5}$$

where $M$ is the total mass of the monomers in the polymer and $r_{cm}$ the center of mass position. In Figure 7 the chain extension for the brushes coated on the inner cylinder and brushes coated on the outer cylinder are shown for $R_{in} = 5, 20$ and $50\sigma$. In the right figures the exponent is shown as a

function of the inner cylinder radius. In Figure 7a the data for grafting density $0.085\sigma^{-2}$ are shown. At low shearrate, the radius of gyration equals the radius of gyration at zero shear. For these shear rates, we are in thermal equilibrium. Only at high shear rates, we observe that $R_g$ increases. The increase can be described by a power law. The exponent of the power law depends on the radius of the cylinders and decreases from 0.28 to 0.22 for the polymers on the inner cylinder and increases from 0.13 to 0.21 for the polymers on the outer cylinder with the increase of the radius. In Figure 7b the data for grafting density $0.170\sigma^{-2}$ are shown. Here the exponent decreases from 0.39 to 0.30 for the inner cylinder and increases from 0.17 to 0.28 for the outer cylinder with the increase of the radius. These changes in the exponent that depend on the cylinder radius are clearly different from the exponents that are observed in the case of flat brushes [27,31] The overview of the exponents in the graphs on the right in Figure 7 show the shift from asymmetrical polymer extension on the lower radii to symmetrical polymer extension for $R \to \infty$. This agrees with the results from Figure 6, where we see an shift in the middle of the polymer overlap from near the inner wall to the center between the walls for higher radii.

In Figure 8 the velocity profiles for brushes grafted on cylinders with small radii ($R_{in} = 5\sigma$, Figure 8a,b and large radii ($R_{in} = 50\sigma$, Figure 8c,d for $\dot\gamma = 1 * 10^{-2}\tau^{-1}$, $\dot\gamma = 6 * 10^{-2}\tau^{-1}$ and $\dot\gamma = 1 * 10^{-1}\tau^{-1}$ are shown. Except for Figure 8b where $\dot\gamma = 1 * 10^{-3}\tau^{-1}$, $\dot\gamma = 1 * 10^{-2}\tau^{-1}$ and $\dot\gamma = 6 * 10^{-2}\tau^{-1}$ are used, as for the high shear rate of $\dot\gamma = 1 * 10^1\tau^{-1}$ the shear stresses at the anchor points become too high and the polymers start to hop over the surface. The velocity profiles indeed confirm the asymmetric response of the brushes on the inner and outer cylinder to the applied shear stress. The velocity for the inner polymers is higher than the outer polymers. For lower grafting densities, Figure 8a,c, we see the velocity profile shorten for the inner polymers at the highest shearrates. This can be attributed to the large difference in chain extension we saw in Figure 7. Where the inner polymers are more compressed than the outer polymers. At high grafting density, Figure 8b,d we see a more even distribution, this is likely due to the high compression of the system.

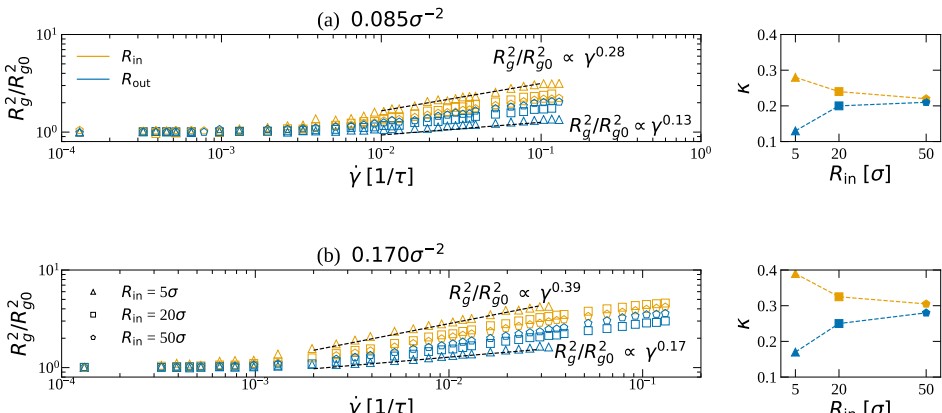

**Figure 7.** The chain extension $R_g^2/R_{g0}^2$ for the brushes grafted on the interior and exterior of the cylinder models for different $R$ under different shear rates. (**a**) The chain extension $R_g^2/R_{g0}^2$ as a function of the shear rate for a grafting density of $\alpha_g = 0.085\,\sigma^{-2}$. The graph on the side shows the exponent kappa of the chain extension for the inner and the outer brush as a function of the cylinder radius. (**b**) The chain extension $R_g^2/R_{g0}^2$ as a function of the shear rate for a grafting density of $\alpha_g = 0.170\,\sigma^{-2}$. The graph on the side shows the exponent kappa of the chain extension for the inner and the outer brush as a function of the cylinder radius.

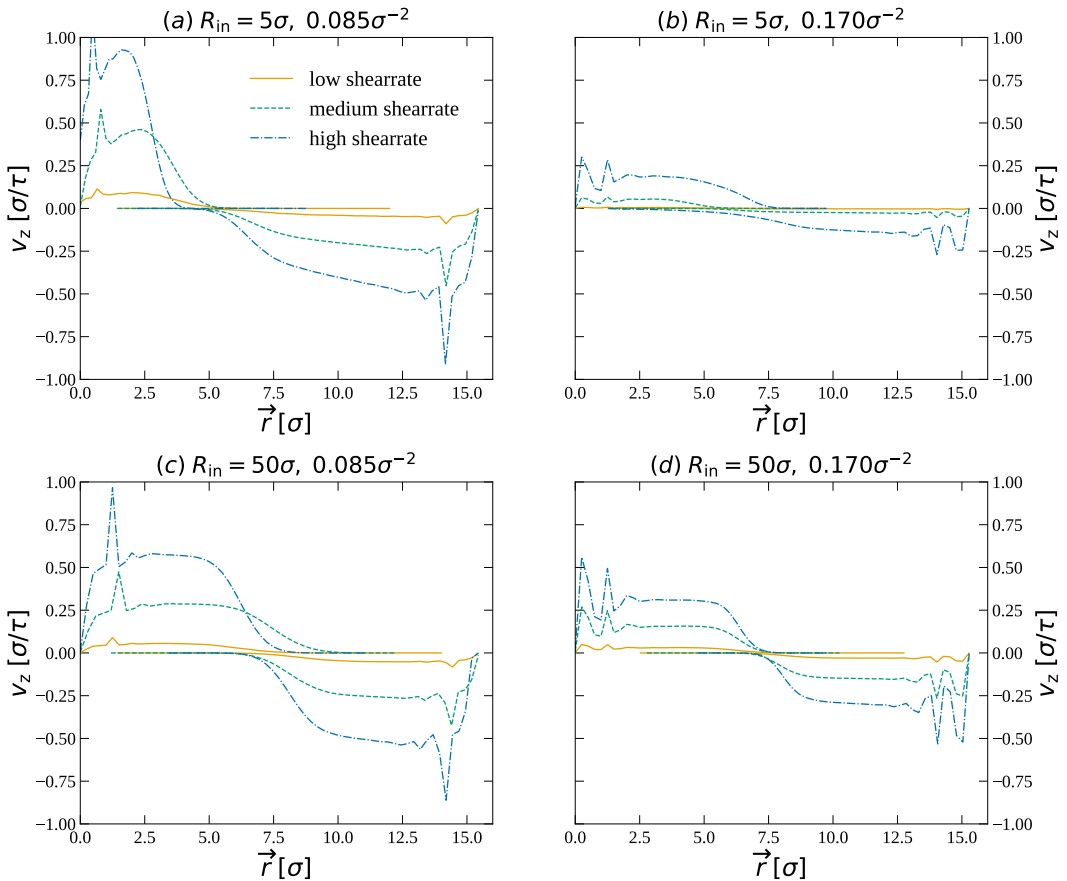

**Figure 8.** The velocity profiles for brushes grafted on the interior and exterior of the cylinder models of different radii $R$ for $\dot{\gamma} = 1 * 10^{-2}\tau^{-1}$, $\dot{\gamma} = 6 * 10^{-2}\tau^{-1}$ and $\dot{\gamma} = 1 * 10^{-1}\tau^{-1}$ are shown. The brushes are grafted on the exterior of the inner wall and the exterior of the outer wall. (**a**) The velocity $v_{\rm in}$ (left curve) and $v_{\rm out}$ (right curve) as a function of the distance from the surface $\overrightarrow{r}$ for a grafting density of $\alpha_{\rm g} = 0.085\ \sigma^{-2}$ and cylinder radius $R_{\rm in} = 5\sigma$. (**b**) The velocity $v_{\rm in}$ (left curve) and $v_{\rm out}$ (right curve) as a function of the distance from the surface $\overrightarrow{r}$ for a grafting density of $\alpha_{\rm g} = 0.170\ \sigma^{-2}$ and cylinder radius $R_{\rm in} = 5\sigma$. (**c**) The velocity $v_{\rm in}$ (left curve) and $v_{\rm out}$ (right curve) as a function of the distance from the surface $\overrightarrow{r}$ for a grafting density of $\alpha_{\rm g} = 0.085\ \sigma^{-2}$ and cylinder radius $R_{\rm in} = 50\sigma$. (**d**) The velocity $v_{\rm in}$ (left curve) and $v_{\rm out}$ (right curve) as a function of the distance from the surface $\overrightarrow{r}$ for a grafting density of $\alpha_{\rm g} = 0.170\ \sigma^{-2}$ and cylinder radius $R_{\rm in} = 50\sigma$.

## 3.2. Friction and Effective Viscosity

Now we turn our attention to the frictional response between brushes grafted in the interior of an outer cylinder and the exterior of an inner cylinder. The inner cylinder is moved over the cylinder axis in the opposite direction of the outer cylinder at velocities $v$ and $-v$ for the inner and outer cylinder, respectively. The resulting friction forces $f_{\rm z}$ as a function of the shear rate $\dot{\gamma}$ are shown in Figure 9. At low shear rates the friction increases linearly with the shear rate. For these shear velocities, the motion of the brushes is slow enough that the polymers are in thermal equilibrium. At higher shear rates the friction force increases sub-linearly with the shear rate via a power law relation $f_{\rm z} \propto \dot{\gamma}^{-\kappa}$, where $\kappa$ is the shear-thinning exponent, which we find to be $\kappa \approx 0.53 - 0.54$. Despite the strong $R$-dependent shear-induced shift in the density profiles and position of the overlap zone (see Figures 5 and 6), the exponent is independent of surface curvature and for all cylinder radii equal to the value observed for bilayer brushes between flat surfaces [27,28,31]. Moreover, upon comparing Figure 9a,b,

one can see that the shear-thinning exponent is also independent of the grafting density of the brush. Overall, the friction forces decrease upon decreasing the cylinder radius. An important reason for this is the decrease in surface area for the smaller cylinder, as we will discuss in more detail below.

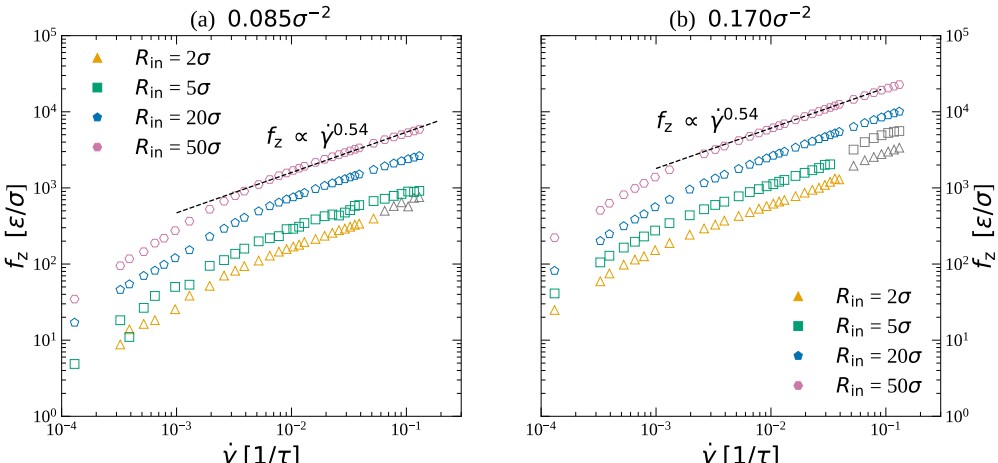

**Figure 9.** The total friction force $f_z$ for the brushes grafted on the interior and exterior of the cylinder models for different $R$ under different shear rates. (**a**) The total friction force $f_z$ as a function of the shear rate for a grafting density of $\alpha_g = 0.085 \, \sigma^{-2}$. All radii follow the same shear-thinning exponent of 0.54 (**b**) The total friction force $f_z$ as a function of the shear rate for a grafting density of $\alpha_g = 0.170 \, \sigma^{-2}$. All radii follow the same shear-thinning exponent of 0.54. The data in gray are discarded due to surface hopping of the polymers.

At high shear rates the shear stresses at the anchor points become too high and the polymers can start to hop over the surface. The data where polymers are suffering from this effect have been colored gray in Figure 9. The onset of polymer-hopping decreases to lower shear rates upon decreasing the cylinder radius. This implies that physically bonded polymers coated on thin cylinders might be more prone to wear and will benefit from techniques to improve the surface bonding of brushes to surfaces that have recently been developed [60,61].

To compare the results for the different cylinder radii in more detail, we can normalize the friction force by the surface areas of the cylinders and extract the effective viscosity $\eta_z$ of the brushes in contact, which is calculated via:

$$\eta_z = \frac{1}{2} \left( \frac{f_{z,\text{in}}}{A_\text{in}} + \frac{f_{z,\text{out}}}{A_\text{out}} \right) \frac{D}{2v} \, , \tag{6}$$

where $f_{z,\text{in}}$ and $f_{z,\text{out}}$ are the forces on the inner and outer cylinder, respectively, and $A_\text{in}$ and $A_\text{out}$ are the surface areas of the inner and outer cylinder, respectively. In Figure 9, we show this effective viscosity as a function of the Weissenberg number $W = \dot{\gamma}/\dot{\gamma}^*$, with $\dot{\gamma}^*$ being the critical shear rate. We determine the critical shear rate via the intersection of the fitted linear response at low shear rates and the fitted power law at high shear rates. The critical shear rate is independent of the cylinder radius. However, it depends on the grafting density. We find that $\dot{\gamma}^* = 0.045\tau^{-1}$ for $\alpha_g = 0.085 \, \sigma^{-2}$ and that $\dot{\gamma}^* = 0.020\tau^{-1}$ for $\alpha_g = 0.170 \, \sigma^{-2}$. These critical shear rates agree with the critical shear rates for brushes of the same grafting densities and wall-wall distances $D$, but then grafted on flat surfaces [27].

It should be noted that the results obtained here do not account for entanglement effects, as the selected chain length is too short for this. However, as friction can only arise from the overlap region, which is typically narrow relative to the total chain length, entanglement effects could only influence the shear response in the case of extremely long chains [28].

Interestingly, Figure 10 shows that the effective viscosity of the brushes is independent of the cylinder radii and they all display the same behavior as a function of the shear rate. This is surprising considering all the changes we observe in the (evolution of the) density profiles (see Section 3.1).

The effective viscosity $\eta_z$ shown in Figure 10 is independent of the shear rate for low shear rates. At these low velocities, the brushes are in thermal equilibrium and behave like a Newtonian liquid. At shear rates above the critical shear rate, we observe shear thinning. In this regime, the effective viscosity decreases with increasing shear rate. The decrease in viscosity can be described by a power law dependency $\eta_z \propto \dot{\gamma}^\kappa$, with $\kappa$ being $-0.46$. Though the effective viscosity increases by approximately a factor 10 upon increasing the grafting density from $\alpha_g = 0.085\,\sigma^{-2}$ to $\alpha_g = 0.170\,\sigma^{-2}$, the shear-thinning exponent is the same for both systems. Moreover, the viscosities for bilayer brushes in a cylindrical geometry (shown in Figure 7) are in almost quantitative agreement with the results reported for brushes between flat surfaces [27,28,31].

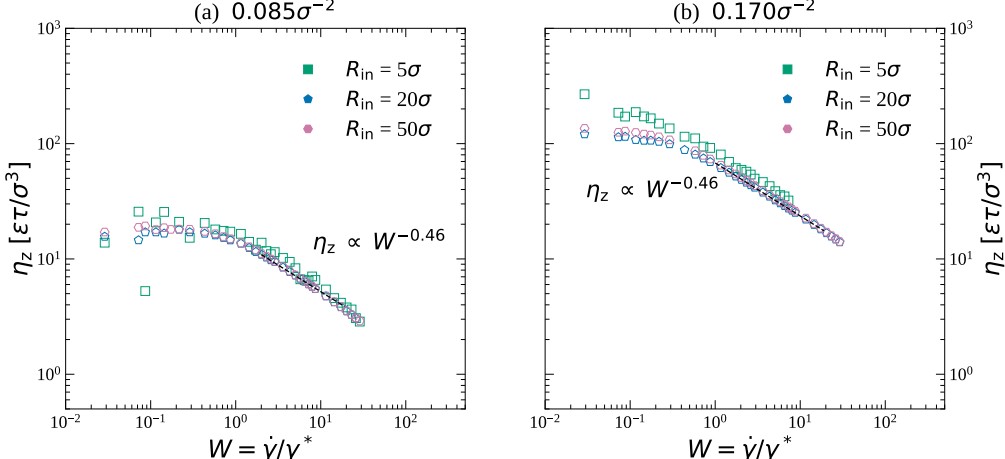

**Figure 10.** The effective viscosity $\eta_z$ for the brushes grafted on the interior and exterior of the cylinder models for different $R$ under different shear rates. (**a**) The effective viscosity $\eta_z$ as a function of the shear rate for a grafting density of $\alpha_g = 0.085\,\sigma^{-2}$. (**b**) The effective viscosity $\eta_z$ as a function of the shear rate for a grafting density of $\alpha_g = 0.170\,\sigma^{-2}$.

Why are the effective viscosities for the brushes between cylinders of different radii very similar when the density profiles (Figures 3 and 5), the shear rate dependent change in the density profiles (Figure 6) as well as the shear rate dependent changes in the tilting angles (Figure 7) are significantly dissimilar? To understand this, we must consider the theory proposed in Refs [27,28,31]. These authors state that the friction between opposing brushes in relative sliding motion is determined by the overlap between the brushes. The effective interaction experienced by the brushes depends on the monomer concentration $c$ multiplied by the overlap length $l$. Equations that have been derived for the overlap length $l$ based on mean field and scaling approximations [62] or via analytical self-consistent field theory [63] are very similar. This implies that the overlap between the brushes is rather insensitive to the exact density profiles and that instead, it is mainly determined by the monomer concentration, which is controlled by the grafting density $\alpha_g$ and the wall-to-wall distance $D$. In Figure 6, we show that the monomer concentration is indeed independent of the radii of the cylinders, but that it strongly depends on the grafting density.

To validate that the overlap between the brushes is indeed insensitive to the exact density profiles, we compute the overlap integral as defined in section 2. The results are shown in Figure 11. The data in Figure 11 show that the overlap integral is approximately the same for the differently curved cylinders and depends on the grafting density alone. For $\alpha_g = 0.085\sigma^{-2}$ it is approximately 0.15 for low velocities and for $\alpha_g = 0.170\sigma^{-2}$ it is approximately 0.5 for low velocities. As expected the overlap between the brushes is constant for $\dot{\gamma} < \dot{\gamma}^*$. For these shear rates, the brushes are in thermal equilibrium. At shear rates $\dot{\gamma} > \dot{\gamma}^*$, the overlap between the brushes decreases, which is at the origin of the shear-thinning behavior we present in Figure 10.

The independence of curvature in the low-shear regime may be explained based on the constant and homogeneous density throughout our systems. As a result of this constant density, the density of the brushes at the point of greatest overlap, where the density profiles intersect, is the same throughout our systems. Furthermore, the average interaction energy of a particle is independent of its radial position. As a first approximation, we may therefore assume the overlap profile to retain the Gaussian shape reported in existing literature [27,31], since the interpenetration is governed by the entropy of chain extension. At higher shear rates, where the shear-thinning occurs, a similar argument may be employed. In this case, the stretching of the polymers in the shear direction produces a sharper interface, but overlap is once again driven by the entropy of the outermost monomers. This translational entropy is balanced against the shear force. Assuming an otherwise symmetrical situation, balance of forces requires the average shear force per particle in the overlap region to be equal for both brushes. As the shear force in the strong-stretching case is directly proportional to the number of particles, this once again leads to the independence of the overlap integral on the system radius. It should be noted, however, that the preceding explanation relies on strong simplifications of the chain stretching effects, neglecting for example the fact that the system radius impacts the difference in chain number between the inner and outer brush, and therefore the shear force per chain. A more formal investigation of these results may be of value.

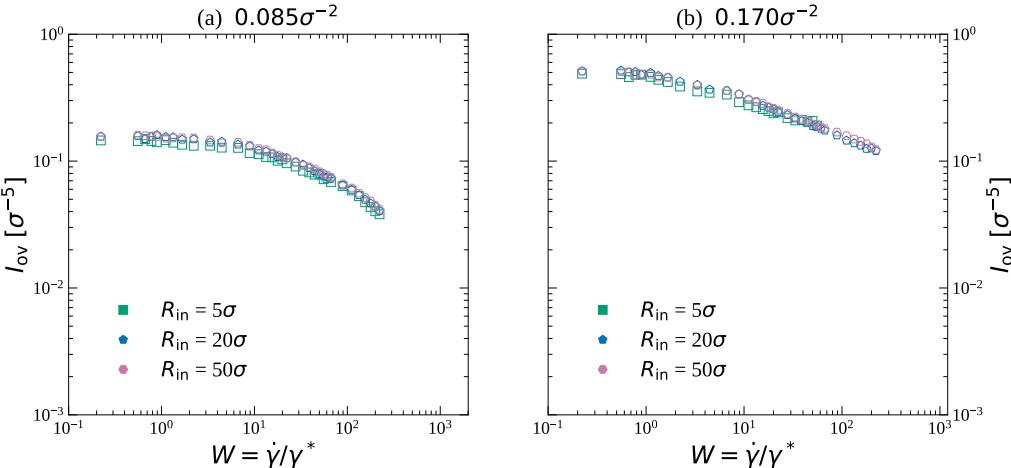

**Figure 11.** The overlap integral $I_{ov}$ for the brushes grafted on the interior and exterior of the cylinder models for different $R$ under different shear rates. (**a**) The overlap integral $I_{ov}$ as a function of the shear rate for a grafting density of $\alpha_g = 0.085\ \sigma^{-2}$. (**b**) The overlap integral $I_{ov}$ as a function of the shear rate for a grafting density of $\alpha_g = 0.170\ \sigma^{-2}$.

## 4. Conclusions

In summary, we have shown that the lubricity of bilayer brushes in cylindrical geometries is independent of the curvature of the cylindrical walls. These results are surprising considering that the density distributions and the positioning of the overlap zone depend strongly on the cylinder radius as well as the shear rate. In equilibrium, brushes on the exterior of cylinders display a shallower density distribution compared to brushes on flat surfaces, while brushes on the interior of cylinder are denser and show a sharper polymer profile. When brushes on the exterior of an inner cylinder are brought into contact with brushes on the interior of an outer cylinder, the overlap zone is shifted off-center towards the inner cylinder. This shift is stronger for smaller cylinder radii. Under shear motion the overlap zone shifts towards the center again.

All observations described above confirm that studies performed on brushes between parallel plates cannot be translated directly to the behavior of brushes in cylindrical geometries. Nevertheless, despite all these differences, the effective viscosity of the brushes confined between cylinders is

independent of the cylinder radius and equal to the viscosity of brushes between parallel plates. The reason for this is that the effective viscosity is determined by the amount of overlap between the brushes. The overlap between brushes is rather insensitive to the exact density distributions and instead depends only on the grafting density and the distance between the (cylinder) walls. We observe indeed that the overlap and the average concentration is constant for all cylinders. Our results imply that research on the lubricating properties of brushes between flat plates can be employed to predict the lubricity of brushes in cylindrical geometries.

**Author Contributions:** Conceptualization, K.J.v.d.W., G.C.R.v.E. and S.d.B.; methodology, K.J.v.d.W., G.C.R.v.E. and S.d.B.; software, K.J.v.d.W. and S.d.B.; validation, K.J.v.d.W., G.C.R.v.E. and S.d.B.; formal analysis, K.J.v.d.W., G.C.R.v.E. and S.d.B.; investigation, K.J.v.d.W., G.C.R.v.E. and S.d.B.; resources, K.J.v.d.W. and S.d.B.; data curation, K.J.v.d.W.; writing—original draft preparation, K.J.v.d.W., G.C.R.v.E. and S.d.B.; writing—review and editing, K.J.v.d.W., G.C.R.v.E. and S.d.B.; visualization, K.J.v.d.W., G.C.R.v.E. and S.d.B.; supervision, S.d.B.; project administration, K.J.v.d.W.; funding acquisition, S.d.B.

**Funding:** The HPC resources were funded by the Netherlands Organisation for Scientific Research (NWO), project number 15987.

**Conflicts of Interest:** The authors declare no conflict of interest.

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
