# Peer review of "Polymer Brush Friction in Cylindrical Geometries"

_lubricants, doi:10.3390/lubricants7100084_

Round 1

Reviewer 1 Report

See the attachment.

Author Response

This paper studies tribological behavior of surface-grafted polymer chains us- ing molecular dynamics simulation technique. The polymer chains are grafted on interior and exterior surfaces of cylindrical geometries of different radii rather than usual flat-surfaces. Two grafting densities have been probed. The smaller cylinder having chains grafted on exterior surfaces are moving inside the larger cylinders having chains grafted on interior surfaces. The results provide insights into the friction mechanism of polymer brushes grafted on cylindrical geometries. 

I recommend the publication of this paper. However, I would like to have clarifications regarding some of the informations in the paper. Details are below. 

We thank the reviewer for the careful assessment of the paper.

The authors are studying polymer brushes on cylindrical surfaces. It would be nice if authors can give examples of real life systems (may be in living systems) with such geometries. It would also be interesting to compare the sizes and ratios of the cylinders and brushes used in the real systems with the one used in this work. 

We added more details and examples of real life systems with such geometries. Polymer brush coated needles can be used to reduce friction and improve needling injection accuracy. Another example would be the nuclear pore complex, which is often modeled as a brush on the inside of a cylinder.

Additionally, we have added a discussion of the sizes and ratios in real systems.

On page 3, the authors discuss about bead size and Kuhn monomer. The authors say ”one bead represents 3-4 monomers”. It would be better to write ”... may represent ....”. The authors have used the example of polyethylene. Polyethylene has a Kuhn size of lk = 1.4 nm [pg 53: table 2.3, Polymer Physic, Rubinstein and Colby, 2008]. So a bead of size σ = 0.5 nm does not represent a Kuhn unit. This parts is a bit confusing and should be rewritten. The unit of velocity v = 7 m/s is also not clear. 

We have extended our explanation and make it more clear that these values have been derived in Kremer, K.; Grest, G.S., J. Chem. Phys. 1990, 92, 5057.

It will also be nice if the authors can provide some velocity profile of brushes under shear on the cylindrical geometries.

We agree with the reviewer and we have added the velocity profiles to the main text. The velocity profiles indeed confirm the highly asymmetric response as observed for the density profiles.

Reviewer 2 Report

It is an interesting article dealing with MD simulation of comb-like materials in cylinder.  The methodology adopted is appropriate and the results were clearly presented.  Only minor revision is required:  on page 7, line 6 from bottom, the authors claimed that '...cannot determine the exact exponent of the powerlaw...'  However, as can be seen from Fig.4, one still can have an exponent.  Certainly it won't be -4.  Please explain why.  If it were not integer, it would be more intriguing.  Certainly, the authors should also make some statements about it.

Author Response

It is an interesting article dealing with MD simulation of comb-like materials in cylinder.  The methodology adopted is appropriate and the results were clearly presented.  Only minor revision is required:  on page 7, line 6 from bottom, the authors claimed that '...cannot determine the exact exponent of the powerlaw...'  However, as can be seen from Fig.4, one still can have an exponent.  Certainly it won't be -4.  Please explain why.  If it were not integer, it would be more intriguing.  Certainly, the authors should also make some statements about it. 

We thank the reviewer for the positive assessment of the paper.

We agree with the reviewer that it would be interesting to obtain a powerlaw exponent for the height of the brushes on the outer cylinder radius. Yet, this is not possible, since the transition to a constant height for R -> infinity has already set in. Therefore, the exponent that we would derive from a fit to the data will have an arbitrary value between -4 and 0 and will be meaningless.

We have adjusted our explanation in the main text to make this more clear.

Reviewer 3 Report

In this manuscript, the authors used molecular simulation to study the lubrication of polymer brushes in cylindrical geometries. Such cylindrical geometry certainly brings a lot of differences to the lubrication behaviors of polymer brushes. While the study is certainly interesting and practically important, I found this manuscript is not suitable for publication in Lubricants at this moment but may be accepted after a revision.

(1) My biggest concern is that the authors did not include solvent in their system. The solvent is essential to the lubrication of polymer brushes in experiments. It has also been shown that the solvent molecules make a big difference in the brush density profile and lubrication properties in MD simulation (Galuschko, Langmuir 2010, 26 (9), 6418-6429). Can the authors comment on how the solvent can affect their current results?

(2) The grafting densities used in this study are pretty high. The high grafting densities lead to the saturation effects and therefore the effective viscosity is independent of the radius of the cylinders. The effective viscosity may depend on the radius of the cylinder at lower grafting density.

(3) The chain length used in this study (N = 30) is too low to observe the entanglement effect.

Author Response

Reviewer 3:

In this manuscript, the authors used molecular simulation to study the lubrication of polymer brushes in cylindrical geometries. Such cylindrical geometry certainly brings a lot of differences to the lubrication behaviors of polymer brushes. While the study is certainly interesting and practically important, I found this manuscript is not suitable for publication in Lubricants at this moment but may be accepted after a revision.

We thank the reviewer for a well-considered and critical assessment of the manuscript.

(1) My biggest concern is that the authors did not include solvent in their system. The solvent is essential to the lubrication of polymer brushes in experiments. It has also been shown that the solvent molecules make a big difference in the brush density profile and lubrication properties in MD simulation (Galuschko, Langmuir 2010, 26 (9), 6418-6429). Can the authors comment on how the solvent can affect their current results?

The differences between explicitly solvated and solvent-free systems are significant indeed, and stem from increased particle densities in the system as well as solvent hydrodynamics. In an explicitly solvated system, both shear and normal forces are generally larger due to the higher particle density. However, as discussed in the aforementioned paper, neither of these differences fundamentally distinguish the steady-state shear response of an explicitly solvated system from that of a solvent-free one, as long as hydrodynamic relations are properly preserved in the latter. In fact, the two cases meet in the limit of high grafting density by necessity. The results obtained by Galuschko et al. are consistent with this. The authors attribute this to their use of a DPD thermostat, which satisfies local conservation of momentum. Instead of a DPD thermostat, we employed a Langevin thermostat operating in the non-shear directions. This has been shown (Pastorino, Phys. Chem. Chem. Phys., 2002, 4, 3008-3015) to produce functionally the same results outside extreme non-equilibrium situations. Because of these former results, we see no reason to expect geometry-specific solvent effects.

We realize that this might not be clear to everyone. Therefore, we have extended our model description to clarify this.

(2) The grafting densities used in this study are pretty high. The high grafting densities lead to the saturation effects and therefore the effective viscosity is independent of the radius of the cylinders. The effective viscosity may depend on the radius of the cylinder at lower grafting density.

We fear that the reviewer might have misinterpreted our grafting desnities. We point out that the grafting densities used fall within a fairly typical range for simulations, and are a selection of those studied in Galuschko et al. and followup work (Spirin, Eur. Phys. J. E 33, 307-311 (2010)). In both these studies, the same scaling approach used in this manuscript results in data collapse onto a single master curve for densities ranging from just above the mushroom-brush transition to well in excess of our highest grafting density. The previously mentioned work by Pastorino et al. also incorporates simulations of brushes up to 4.9 times the critical grafting density as a benchmark of the hydrodynamic behaviour of different thermostats. Similar or higher grafting densities abound in literature (e.g. [Adiga and Brenner, J. Funct. Biomater. 2012, 3, 239-256], [De Beer and Muser, Soft Matter, 2013, 9, 723], [Dimitrov et al., J. Chem. Phys. 127, 084905 (2007)]) with no reports of density-related artefacts. Considering the abundance of prior literature using similar parameters, and particularly the fact that the scaling obtained matches previous results and theory, we do not believe that such effects play a role here.

We believe the presentation of the grafting density in the reduced units of the simulation system may have led to confusion on this point. 

(3) The chain length used in this study (N = 30) is too low to observe the entanglement effect.

The existing theory we intend to expand upon in this manuscript is based on brushes that are both highly compressed and non-entangled. For comparability of results, we have limited ourselves to the same system parameters. We recognise that this limits the applicability of the results presented, but do not believe it diminishes their relevance. Moreover, the aforementioned work by Spirin et al. provides a compelling qualitative argument as to why entanglement effects should not be expected in these systems. As the shear response ultimately originates in the overlap region, entanglements within the individual brushes cannot influence the results. Meanwhile, entanglements between the two brushes can only occur within the overlap region, which is typically narrow. Hence, no entanglement effects are expected up to very large chain lengths. 

We have adjusted our model description to acknowledge this fact, and added an explanation of the argument discussed above.